# PAMAM-Calix-Dendrimers: Second Generation Synthesis, Fluorescent Properties and Catecholamines Binding

**DOI:** 10.3390/pharmaceutics14122748

**Published:** 2022-12-08

**Authors:** Olga Mostovaya, Igor Shiabiev, Dmitry Pysin, Alesia Stanavaya, Viktar Abashkin, Dzmitry Shcharbin, Pavel Padnya, Ivan Stoikov

**Affiliations:** 1A.M. Butlerov Chemical Institute, Kazan Federal University, 18 Kremlevskaya Street, Kazan 420008, Russia; 2Institute of Biophysics and Cell Engineering of NASB, 27 Akademicheskaya Street, 220072 Minsk, Belarus; 3Federal Center for Toxicological, Radiation and Biological Safety, 2 Nauchny Gorodok Street, Kazan 420075, Russia

**Keywords:** thiacalixarene, catecholamines, dendrimer, PAMAM, fluorescent properties, dopamine, *L*-adrenaline, *L*-noradrenaline

## Abstract

A convenient method for the synthesis of the second generation of PAMAM dendrimers based on a *p-tert*-butylthiacalix[4]arene core in *cone*, *partial cone* and *1,3-alternate* conformations was developed. Unusual fluorescence of the obtained PAMAM-calix-dendrimers has been found and explained. The binding ability of the synthesized dendrimers toward catecholamines (dopamine, *L*-adrenaline and *L*-noradrenaline) was shown by UV-Vis, fluorescence, 1D and 2D NMR spectroscopy and the binding constants (logK_a_ 3.85–4.74) calculated. As was shown, the PAMAM-calix-dendrimers bind catecholamines by the internal cavities. All the studied hormones were most efficiently bound by the dendrimers bearing a macrocyclic core in *1,3-alternate* conformation. The size of the formed supramolecular systems of dendrimer/catecholamine was established by the DLS method. A decrease in hemolytic activity of the PAMAM-calix-dendrimers with an increase in the generation number of a dendrimer was shown for the dendrimers with a core in *1,3-alternate* conformation. The prospects for the use of the synthesized dendrimers with the macrocyclic core as drug delivery agents were discussed.

## 1. Introduction

In recent decades, catecholamines have gained great attention from scientists working in various scientific areas. This is because of their significant role in many physiological processes in the human body. Catecholamines, i.e., dopamine, *L*-adrenaline and *L*-noradrenaline, are neurotransmitters and play an important role in functioning of the nervous system [1]. The relationship between the functioning of the nervous and immune systems and the importance of these compounds for immunomodulation have also been revealed [2]. An excess or deficiency of these compounds in the body leads to many diseases including neurodegenerative disorders [3,4]. Significant increase in the life expectancy has already led to an increase in neurodegenerative diseases (about 10 million new dementia cases per year) [5,6]. The application of catecholamines in anesthetic practice as vasopressors and inotropes is also well known [7]. Thus, the importance and relevance of the design of the compounds capable of binding catecholamines becomes obvious. The reversible binding of catecholamines may be useful for the creation of sustained-release formulations. The synthesis of such compounds opens new prospects both in the diagnostics and treatment of several severe diseases. Moreover, such compounds may be useful in the development of new drug delivery systems.

Dendrimers, i.e., polymeric hyperbranched synthetic molecules with regular structures, have been well proven for binding of various compounds. Compared to common polymers, these monodisperse and symmetrical molecules have a defined structure [8]. Variation of numerous terminal functional groups (neutral or positively/negatively charged) makes it possible to control the molecule hydrophilicity/hydrophobicity. This offers the ability of dendrimers to bind various substrates and transport them in various media [9,10,11].

The nanometer size of dendrimer molecules comparable to that of the drug delivery systems facilitates their application in medical technologies [12]. They can form 10–500 nm nanoparticles, which well penetrate the cells and promote the immune response [13,14,15]. Such aggregates have been proposed as promising drug delivery systems due to the ability to covalently and noncovalently bind guest molecules and penetrate the blood–brain barrier [16,17]. Application of these dendrimers is also relevant in the treatment of some brain diseases, e.g., oncological, neurodegenerative and ischemic disorders of cerebral circulation.

Dendrimers bind substrates in various simultaneous ways, e.g., by internal cavities encapsulation, peripheral attachment, internal cavities encapsulation and peripheral attachment [9]. The dendrimer platform is ideally suited for binding drug molecules. These molecules can both be physically distributed inside the cavities of dendrimers and can bind to it through covalent bonds and a whole range of noncovalent interactions (electrostatic interactions, hydrogen bonding and van der Waals forces) [12,18]. High-generation dendrimers are mostly used in encapsulation and subsequent drug delivery [15]. However, such compounds frequently exert high cytotoxicity [19,20]. Significant cost of high dendrimer generations caused by the large number of repetitive synthetic procedures is another problem of their application [21]. For this reason, low-generation dendrimers capable of effective binding of target substrates have received an advantage in possible future application. In contrast, the number of binding centers in such molecules is rather low. This can adversely affect the efficiency of appropriate interactions with guest molecules.

First and second generations of dendrimers with the thiacalix[4]arene central core have been recently obtained to compare the binding efficiency depending on the number of binding centers. The choice of this macrocycle as a dendrimer core is explained by its attractive properties [22,23,24], including the possibility of fixing the platform in the desired conformation and the ability to preorganize the fragments participating in target bindings. The presence of several functional groups determines the possibility of the multivalency effect in such binding [25,26]. Finally, the size of the macrocyclic platform meets the size of low-generation PAMAM dendrimers [15].

This work includes the development of synthetic methods for second-generation dendrimers based on the thiacalix[4]arene and their application for catecholamine binding. The relationship between the dendrimer generation and the catecholamine binding efficiency has been established, and the structure of the resulting complexes was proven by a series of physical methods. It is interesting that there are practically no examples in the literature of the use of PAMAM dendrimers for catecholamine binding. A dendrimer containing β-cyclodextrin fragments as terminal groups has been obtained quite recently [27]. The authors propose it as a dopamine sensor but do not address the issues raised in the current manuscript.

## 2. Materials and Methods

### 2.1. General Experimental Information

More details on the equipment, methods of confirmation and establishment of the compounds structure are described in the Appendix A.

Most chemicals (dopamine hydrochloride, *L*-adrenaline hydrochloride and *L*-noradrenaline hydrochloride, first and second generation PAMAM dendrimers with ethylenediamine core) were purchased from Aldrich. KH_2_PO_4_ (HPLC grade) was purchased from Fisher Scientific and used as received without additional purification. Organic solvents were purified in accordance with standard procedures. Deionized water with resistivity > 18.0 MΩ cm (Millipore-Q) was used for the preparation of the solutions.

The PAMAM-calix-dendrimers **G1** were synthesized by the previously described protocol [28], and **G0-monomer**, by that described in [29].

### 2.2. General Procedure for the Synthesis of the Compounds G1.5

The solution (10% by the weight) of the **G1** (1 g, 0.44 mmol) in methanol was added dropwise to an ice-cooled (0 °C) methyl acrylate (1.3 mL, 14.08 mmol) dissolved in 3 mL of methanol. It was kept in the ice bath for another 2 h. Then, the reaction mixture was stirred at room temperature for 36 h. Afterward, the remaining methyl acrylate and the solvent were removed on a rotary evaporator. The residue was dried under reduced pressure.

#### 2.2.1. 5,11,17,23-Tetra-*tert*-butyl-25,26,27,28-tetrakis[*N*-(6-(*N*,*N*-di(*N*-(2-(*N*,*N*-di(methoxycarbonylethyl)amino)ethyl)carbamoylethyl)amino)hexyl)carbamoylmethoxy]-2,8,14,20-tetrathiacalix[4]arene, [**G1.5-cone**]. Viscous Yellowish Oil, Yield: 1.52 g (95%)

^1^H NMR (CD_3_OD, δ, ppm, *J*/Hz): 1.14 (s, 36H, (CH_3_)_3_C), 1.35 (m, 16H, C(O)NHCH_2_CH_2_CH_2_CH_2_), 1.50 (m, 8H, CH_2_CH_2_CH_2_N), 1.60 (m, 8H, C(O)NHCH_2_CH_2_CH_2_), 2.38 (t, 16H, NCH_2_CH_2_C(O)NH, ^3^*J*_HH_ = 7.0), 2.41–2.50 (m, 40H, CH_2_CH_2_CH_2_N, NCH_2_CH_2_C(O)OCH_3_), 2.55 (t, 16H, NHCH_2_CH_2_N, ^3^*J*_HH_ = 6.3), 2.70–2.83 (m, 48H, NCH_2_CH_2_C(O)), 3.25 (t, 16H, NHCH_2_CH_2_N, ^3^*J*_HH_ = 6.3), 3.33 (m, 8H, NHCH_2_CH_2_CH_2_), 3.66 (s, 48H, OCH_3_), 4.89 (s, 8H, OCH_2_C(O)), 7.45 (s, 8H, ArH).

^13^C NMR (CD_3_OD, δ, ppm): 27.92, 28.15, 28.45, 30.73, 31.66, 33.58, 34.37, 35.25, 38.46, 40.38, 50.49, 50.85, 52.22, 53.77, 54.48, 75.36, 129.80, 136.09, 148.85, 159.31, 170.49, 174.69, 174.70.

FTIR ATR (ν, cm^−1^): 3318 (N-H), 1734 (C=O), 1648 (C(O)NH, amide I), 1536 (C(O)NH, amide II), 1093 (C_Ph_OCH_2_).

#### 2.2.2. 5,11,17,23-Tetra-*tert*-butyl-25,26,27,28-tetrakis[*N*-(6-(*N*,*N*-di(*N*-(2-(*N*,*N*-di(methoxycarbonylethyl)amino)ethyl)carbamoylethyl)amino)hexyl)carbamoylmethoxy]-2,8,14,20-tetrathiacalix[4]arene, [**G1.5-paco**]. Viscous Yellowish Oil, Yield: 1.57 g (98%)

^1^H NMR (CD_3_OD, δ, ppm, *J*/Hz): 1.09 (s, 18H, (CH_3_)_3_C), 1.30–1.72 (m, 32H, C(O)NHCH_2_CH_2_CH_2_CH_2_, C(O)NHCH_2_CH_2_CH_2_CH_2_, C(O)NHCH_2_CH_2_CH_2_CH_2_, CH_2_CH_2_N), 1.35 (s, 18H, (CH_3_)_3_C), 2.32–2.40 (m, 16H, NCH_2_CH_2_C(O)NH), 2.41–2.51 (m, 40H, CH_2_CH_2_CH_2_N, NCH_2_CH_2_C(O)OCH_3_), 2.52–2.60 (m, 16H, NHCH_2_CH_2_N), 2.71–2.85 (m, 48H, NCH_2_CH_2_C(O)), 3.14 (m, 2H, C(O)NHCH_2_), 3.25 (m, 16H, NHCH_2_CH_2_N), 3.28–3.42 (m, 6H, C(O)NHCH_2_), 3.66 (s, 48H, OCH_3_), 4.25 (d, 2H, OCH_2_C(O), ^2^*J*_HH_=13.7), 4.60 (s, 2H, OCH_2_C(O)), 4.86 (s, 2H, OCH_2_C(O)), 5.01 (d, 2H, OCH_2_CONH, ^2^*J*_HH_=13.7), 7.12 (d, 2H, ArH, ^4^*J*_HH_=2.6), 7.61 (d, 2H, ArH, ^4^*J*_HH_=2.6), 7.66 (s, 2H, ArH), 7.84 (s, 2H, ArH).

^13^C NMR (CD_3_OD, δ, ppm): 27.93, 28.07, 28.13, 28.32, 28.43, 28.51, 30.59, 30.73, 31.71, 31.81, 31.90, 33.60, 34.38, 38.46, 40.02, 40.37, 40.43, 50.50, 50.83, 52.21, 53.78, 54.42, 54.47, 54.60, 70.77, 74.12, 74.57, 127.98, 129.48, 130.06, 134.68, 135.56, 135.98, 137.53, 147.18, 147.61, 148.86, 157.91, 159.40, 160.87, 170.10, 170.53, 170.76, 174.72.

FTIR ATR (ν, cm^−1^): 3300 (N-H), 1734 (C=O), 1648 (C(O)NH, amide I), 1534 (C(O)NH, amide II), 1088 (C_Ph_OCH_2_).

#### 2.2.3. 5,11,17,23-Tetra-*tert*-butyl-25,26,27,28-tetrakis[*N*-(6-(*N*,*N*-di(*N*-(2-(*N*,*N*-di(methoxycarbonylethyl)amino)ethyl)carbamoylethyl)amino)hexyl)carbamoylmethoxy]-2,8,14,20-tetrathiacalix[4]arene, [**G1.5-alt**]. Viscous Yellowish Oil, Yield: 1.58 g (99%)

^1^H NMR (CD_3_OD, δ, ppm, *J*/Hz): 1.28 (s, 36H, (CH_3_)_3_C), 1.35 (m, 16H, C(O)NHCH_2_CH_2_CH_2_CH_2_), 1.45–1.62 (m, 16H, CH_2_CH_2_CH_2_N, C(O)NHCH_2_CH_2_CH_2_CH_2_), 2.38 (t, 16H, NCH_2_CH_2_C(O)NH, ^3^*J*_HH_ = 7.0), 2.42–2.51 (m, 40H, CH_2_CH_2_CH_2_N, NCH_2_CH_2_C(O)OCH_3_), 2.55 (t, 16H, NHCH_2_CH_2_N, ^3^*J*_HH_ = 6.3), 2.71–2.82 (m, 48H, NCH_2_CH_2_C(O)), 3.18–3.29 (m, 24H, NHCH_2_), 3.66 (s, 48H, OCH_3_), 4.12 (s, 8H, OCH_2_CO), 7.60 (s, 8H, ArH).

^13^C NMR (CD_3_OD, δ, ppm): 27.91, 28.19, 28.37, 30.74, 31.74, 33.59, 34.38, 35.36, 38.46, 40.72, 50.49, 50.82, 52.20, 53.77, 54.47, 71.59, 129.18, 133.30, 148.91, 157.87, 169.89, 174.67, 174.69.

FTIR ATR (ν, cm^−1^): 3318 (N-H), 1733 (C=O), 1648 (C(O)NH, amide I), 1533 (C(O)NH, amide II), 1086 (C_Ph_OCH_2_).

### 2.3. General Procedure for the Synthesis of the Compounds G2

The solution (10% by the weight) of the **G1.5** (1 g, 0.275 mmol) in methanol was added dropwise to an ice-cooled (0 °C) solution of ethylenediamine (5.9 mL, 88 mmol) in 6 mL of methanol. It was kept in the ice bath for another 2 h. Then, the reaction mixture was stirred at room temperature for 80 h. Afterward, the solvent was removed on a rotary evaporator, and the excess of ethylenediamine was removed by azeotropic distillation with the mixture methanol: toluene (1:9). Then, the remaining toluene was removed by distillation with methanol. The residue was dried under reduced pressure.

#### 2.3.1. 5,11,17,23-Tetra-*tert*-butyl-25,26,27,28-tetrakis[*N*-(6-(*N*,*N*-di(*N*-(2-(*N*,*N*-di(*N*-(2-aminoethyl)carbamoylethyl)amino)ethyl)carbamoylethyl)amino)hexyl)carbamoylmethoxy]-2,8,14,20-tetrathiacalix[4]arene, [**G2-cone**]. White Solid Foam, m.p. 68 °C, Yield: 1.01 g (90%)

^1^H NMR (CD_3_OD, δ, ppm, *J*/Hz): 1.14 (s, 36H, (CH_3_)_3_C), 1.35 (m, 16H, C(O)NHCH_2_CH_2_CH_2_CH_2_), 1.49 (m, 8H, CH_2_CH_2_CH_2_N), 1.59 (m, 8H, C(O)NHCH_2_CH_2_CH_2_CH_2_), 2.30–2.42 (m, 48H, NCH_2_CH_2_C(O)), 2.44–2.51 (m, 8H, CH_2_CH_2_CH_2_N), 2.57 (m, 16H, NHCH_2_CH_2_N), 2.72 (t, 32H, NHCH_2_CH_2_NH_2_, ^3^*J*_HH_=6.3), 2.75–2.84 (m, 48H, NCH_2_CH_2_C(O)), 3.25 (t, 16H, NHCH_2_CH_2_N, ^3^*J*_HH_ = 6.3), 3.33 (m, 8H, NHCH_2_CH_2_CH_2_), 4.12 (s, 8H, OCH_2_C(O)), 7.45 (s, 8H, ArH).

^13^C NMR (CD_3_OD, δ, ppm): 27.88, 28.17, 28.46, 30.74, 31.67, 34.38, 34.79, 38.63, 40.40, 41.99, 42.79, 50.83, 51.17, 53.50, 54.48, 75.39, 129.81, 136.07, 148.88, 159.37, 164.13, 170.54, 174.78, 175.19.

FTIR ATR (ν, cm^−1^): 3278 (N-H), 3061 (N-H), 1635 (C(O)NH, amide I), 1539 (C(O)NH, amide II), 1094 (C_Ph_OCH_2_).

ESI, Calculated [M + 8H + Na]^9+^ *m/z* = 457.3007, [M + 11H + Na]^12+^ *m/z* = 343.2273, [M + 17H + Na]^18+^ *m/z* = 229.1540. Found [M + 8H + Na]^9+^ *m/z* = 457.3228, [M + 11H + Na]^12+^ *m/z* = 343.2446, [M + 17H + Na]^18+^ *m/z* = 229.1657.

#### 2.3.2. 5,11,17,23-Tetra-*tert*-butyl-25,26,27,28-tetrakis[*N*-(6-(*N*,*N*-di(N-(2-(*N*,*N*-di(*N*-(2-aminoethyl)carbamoylethyl)amino)ethyl)carbamoylethyl)amino)hexyl)carbamoylmethoxy]-2,8,14,20-tetrathiacalix[4]arene, [**G2-paco**]. White Solid Foam, m.p. 69 °C, Yield: 0.96 g (85%)

^1^H NMR (CD_3_OD, δ, ppm, *J*/Hz): 1.09 (s, 18H, (CH_3_)_3_C), 1.20–1.73 (m, 32H, C(O)NHCH_2_CH_2_CH_2_CH_2_, C(O)NHCH_2_CH_2_CH_2_CH_2_, C(O)NHCH_2_CH_2_CH_2_CH_2_, CH_2_CH_2_N), 1.35 (s, 18H, (CH_3_)_3_C), 2.28–2.43 (m, 48H, NCH_2_CH_2_C(O)), 2.47 (m, 8H, NCH_2_CH_2_CH_2_), 2.57 (m, 16H, NHCH_2_CH_2_N), 2.67–2.90 (m, 80H, NHCH_2_CH_2_NH_2_, NCH_2_CH_2_CO), 3.14 (m, 2H, C(O)NHCH_2_), 3.25 (m, 48H, NHCH_2_CH_2_N), 3.28–3.42 (m, 6H, C(O)NHCH_2_), 4.26 (br.s, 2H, OCH_2_C(O)), 4.60 (s, 2H, OCH_2_C(O)), 4.84 (s, 2H, OCH_2_C(O)), 4.98 (br.s, 2H, OCH_2_C(O)), 7.12 (s, 2H, ArH), 7.62 (s, 2H, ArH), 7.66 (s, 2H, ArH), 7.84 (s, 2H, ArH).

^13^C NMR (CD_3_OD, δ, ppm): 27.90, 28.16, 28.32, 28.44, 30.59, 30.75, 31.73, 31.82, 31.93, 34.46, 34.85, 35.21, 38.66, 40.06, 40.13, 40.42, 42.05, 42.95, 50.85, 51.21, 53.56, 54.52, 54.65, 70.87, 74.18, 74.55, 128.02, 129.53, 129.97, 134.73, 135.62, 135.99, 137.51, 147.64, 148.86, 157.91, 159.41, 160.91, 170.12, 170.48, 170.82, 174.84, 175.19.

FTIR ATR (ν, cm^−1^): 3279 (N-H), 3061 (N-H), 1635 (C(O)NH, amide I), 1539 (C(O)NH, amide II), 1090 (C_Ph_OCH_2_).

ESI, Calculated [M + 8H + Na]^9+^ *m/z* = 457.3007, [M + 12H]^12+^ *m/z* = 341.3119, [M + 17H + Na]^18+^ *m/z* = 229.1540. Found [M + 8H + Na]^9+^ *m/z* = 457.3228, [M + 12H]^12+^ *m/z* = 341.2290, [M + 17H + Na]^18+^ *m/z* = 229.1658.

#### 2.3.3. 5,11,17,23-Tetra-*tert*-butyl-25,26,27,28-tetrakis[*N*-(6-(*N*,*N*-di(*N*-(2-(*N*,*N*-di(*N*-(2-aminoethyl)carbamoylethyl)amino)ethyl)carbamoylethyl)amino)hexyl)carbamoylmethoxy]-2,8,14,20-tetrathiacalix[4]arene [**G2-alt**]. White Solid Foam, m.p. 75 °C, Yield: 0.99 g (88%)

^1^H NMR (CD_3_OD, δ, ppm, *J*/Hz): 1.28 (s, 36H, (CH_3_)_3_C), 1.34 (m, 16H, C(O)NHCH_2_CH_2_CH_2_CH_2_), 1.42–1.61 (m, 16H, CH_2_CH_2_CH_2_N, C(O)NHCH_2_CH_2_CH_2_CH_2_), 2.31–2.42 (m, 48H, NCH_2_CH_2_C(O)), 2.44–2.52 (m, 8H, CH_2_CH_2_CH_2_N), 2.57 (m, 16H, NHCH_2_CH_2_N), 2.72 (t, 32H, NHCH_2_CH_2_NH_2_, ^3^*J*_HH_=6.3), 2.75–2.84 (m, 48H, NCH_2_CH_2_C(O)), 3.17–3.29 (m, 54H, NHCH_2_), 4.12 (br.s, 8H, OCH_2_C(O)), 7.60 (s, 8H, ArH).

^13^C NMR (CD_3_OD, δ, ppm): 27.85, 28.22, 28.39, 30.77, 31.76, 34.37, 34.79, 38.63, 40.72, 42.03, 42.95, 50.80, 51.16, 53.50, 54.46, 71.53, 129.21, 133.20, 148.86, 157.88, 164.13, 169.89, 174.79, 175.16.

FTIR ATR (ν, cm^−1^): 3282 (N-H), 3061 (N-H), 1636 (C(O)NH, amide I), 1539 (C(O)NH, amide II), 1086 (C_Ph_OCH_2_).

ESI, Calculated [M + 8H + Na]^9+^ *m/z* = 457.3007, [M + 11H + Na]^12+^ *m/z* = 343.2273. Found [M + 8H + Na]^9+^ *m/z* = 457.3231, [M + 11H + Na]^12+^ *m/z* = 343.2446.

### 2.4. Procedure for the Synthesis of the Compound G0.5-Monomer

Methyl acrylate (2.08 mL, 23.2 mmol) was added to the solution of the **G0-monomer** (1.78 g, 5.8 mmol) in methanol (25 mL). The reaction mixture was stirred at room temperature for 12 h. Afterward, the remaining methyl acrylate and the solvent were removed on a rotary evaporator. The residue was dried under reduced pressure.

#### *N*-(6-(*N*,*N*-di(methoxycarbonylethyl)amino)hexyl)-2-(4-(*tert*-butyl)phenoxy)acetamide, [**G0.5-monomer**]. Viscous Yellowish Oil, Yield: 2.70 g (97%)

^1^H NMR (CDCl_3_, δ, ppm, *J*/Hz): 1.24–1.33 (m, 13H, (CH_3_)_3_C, C(O)NHCH_2_CH_2_CH_2_CH_2_), 1.40 (m, 2H, CH_2_CH_2_CH_2_N), 1.53 (m, 2H, C(O)NHCH_2_CH_2_CH_2_), 2.37 (m, 2H, CH_2_CH_2_CH_2_N), 2.42 (t, 4H, NCH_2_CH_2_C(O), ^3^*J*_HH_ = 7.2), 2.74 (t, 4H, NCH_2_CH_2_C(O), ^3^*J*_HH_ = 7.2), 3.32 (m, 2H, NHCH_2_), 3.65 (s, 6H, OCH_3_), 4.46 (s, 2H, OCH_2_C(O)), 6.64 (br.t, 1H, NH), 6.85 (d of AB system, 2H, ArH, ^4^*J*_HH_ = 8.7), 7.32 (d of AB system, 2H, ArH, ^4^*J*_HH_ = 8.7).

^13^C NMR (CDCl_3_, δ, ppm): 26.41, 26.70, 26.85, 29.57, 31.55, 31.95, 38.88, 38.98, 49.14, 51.82, 53.59, 67.55, 114.23, 126.65, 144.96, 155.08, 168.46, 172.73.

FTIR ATR (ν, cm^−1^): 3317 (N-H), 1735 (C=O), 1663 (C(O)NH, amide I), 1538 (C(O)NH, amide II), 1091 (C_Ph_OCH_2_).

ESI, Calculated [M + H]^1+^ *m/z* = 479.3116. Found [M + H]^1+^ *m/z* = 479.3169.

### 2.5. Procedure for the Synthesis of the Compound G1-Monomer

The solution (10% by the weight) of the **G0.5-monomer** (0.7 g, 1.46 mmol) in methanol was added dropwise to an ice-cooled (0 °C) solution of ethylenediamine (3.9 mL, 58.50 mmol) in 4 mL of methanol. It was kept in the ice bath for another 2 h. Then, the reaction mixture was stirred at room temperature for 80 h. Afterward, the solvent was removed on a rotary evaporator, and the excess of ethylenediamine was removed by azeotropic distillation with the mixture methanol: toluene (1:9). Then, the remaining toluene was removed by distillation with methanol. The residue was dried under reduced pressure.

#### *N*-(6-(*N*,*N*-di(*N*-(2-aminoethyl)carbamoylethyl)amino)hexyl)-2-(4-(*tert*-butyl)phenoxy)acetamide, [**G1-monomer**]. Viscous Yellowish Oil, Yield: 0.76 g (97%)

^1^H NMR (CHCl_3_, δ, ppm, *J*/Hz): 1.24–1.33 (m, 13H, (CH_3_)_3_C, C(O)NHCH_2_CH_2_CH_2_CH_2_), 1.43 (m, 2H, CH_2_CH_2_CH_2_N), 1.53 (m, 2H, C(O)NHCH_2_CH_2_CH_2_), 2.01 (br.s, 4H, NH_2_), 2.30–2.43 (m, 6H, NCH_2_CH_2_C(O), CH_2_CH_2_CH_2_N), 2.71 (m, 4H, NHCH_2_CH_2_NH_2_), 2.81 (m, 4H, NCH_2_CH_2_C(O)), 3.23–3.37 (m, 6H, NHCH_2_), 4.45 (s, 2H, OCH_2_C(O)), 6.73 (br.t, 1H, NH), 6.85 (d of AB system, 2H, ArH, ^4^*J*_HH_ = 8.7), 7.32 (d of AB system, 2H, ArH, ^4^*J*_HH_ = 8.7), 7.54 (br.t, 2H, NH).

^13^C NMR (CDCl_3_, δ, ppm): 26.29, 26.68, 27.03, 29.53, 31.57, 34.05, 38.90, 39.00, 41.19, 41.24, 50.11, 53.52, 67.57, 114.28, 126.68, 145.05, 155.08, 168.69, 173.23.

FTIR ATR (ν, cm^−1^): 3287 (N-H), 3073 (N-H), 1646 (C(O)NH, amide I), 1542 (C(O)NH, amide II), 1060 (C_Ph_OCH_2_).

ESI, Calculated [M + H]^1+^ *m/z* = 535.3966, [M + 2H]^2+^ *m/z* = 268.2020. Found [M + H]^1+^ *m/z* = 535.3998, [M + 2H]^2+^ *m/z* = 268.2027.

### 2.6. Preparation of the Compounds G1∙HCl, G2∙HCl and G1-Monomer∙HCl

The compounds **G1**, **G2** and **G1-monomer** were converted into the hydrochloride form. For this purpose, 5 mL of the cooled 17 µM methanol solution of **G1** was mixed with 41 µL (0.51 mmol) of concentrated HCl; 17 µM methanol solution of **G2** was mixed with 96 µL (1.19 mmol) of conc. HCl; and 5 mL of 0.19 mM methanol solution of **G1-monomer** was mixed with 0.11 mL (1.4 mmol) of conc. HCl. After 5 min. of stirring, the solvent was completely removed and the residue dried under reduced pressure.

### 2.7. Study of the PAMAM-Calix-Dendrimers Emission Properties

Fluorescence spectra were recorded on the Fluorolog 3 luminescent spectrometer (Horiba Jobin Yvon, Longjumeau, France). For the **G1** and **G2** dendrimers, the excitation wavelength was selected for spectral properties study at 280 nm, the emission scan range was 300–500 nm, excitation and emission slits were 7 nm and the concentration of the compounds was 50 µM; for the **G1-monomer**, the excitation wavelength was selected at 280 nm, the emission scan range was 300–500 nm, excitation and emission slits were 6 nm and the concentration of the compound was 200 µM; for the **G1-PAMAM** and **G2-PAMAM**, the excitation wavelength was selected at 280 nm, the emission scan range was 300–500 nm, excitation and emission slits were 7 nm and the concentration of the compounds was 50 µM; spectra were recorded at 293 K in methanol and deionized water. Comparison of fluorescence spectra of the **G1-monomer** (200 µM) with the **G1** (50 µM) in water was carried out at a slits of 6 nm (excitation wavelength 280 nm).

### 2.8. Study of Hemolysis Activity

For determination of hemolytic activity, the solutions of **G1-alt** (20 μM) or **G2-alt** (10 μM) in phosphate buffer were added to the erythrocytes (at 2% hematocrit). The samples were incubated at 37 °C for 3 and 24 h, centrifuged (3000× *g*, 10 min, 4 °C), and their absorbance was measured at 540 nm using the spectrophotometer Jasco V-630 (Jasco, Tokyo, Japan). The percentage of hemolysis was calculated using the following formula:(1)$$H\left(\%\right)=\frac{A_{540}}{A_{\mathrm{water}}}\times 100\%$$
where *A_540_* is the sample absorption, and *A_water_*, the RBC positive control in water (100% release of hemoglobin). Data were presented as a percentage of hemolysis, mean ± SD, n = 3.

## 3. Results and Discussion

### 3.1. Synthesis of the Second Generation of Poly(Amidoamine) Dendrimers Based on p-Tert-Butylthiacalixarene (PAMAM-Calix-Dendrimers) in Different Conformations

Previously, we developed a convenient method for the synthesis of the first generation (**G1**) of the PAMAM dendrimers based on *p-tert*-butylthiacalix[4]arene (PAMAM-calix-dendrimers) [28]. This method allows us to obtain dendrimers with the macrocyclic core in all available conformations, i.e., *cone*, *partial cone* and *1,3-alternate*, with high yields. This work became a starting point for the synthesis of the second generation (**G2**) dendrimers. It is well known [30] that the number of reaction centers and probability of the side reactions (*retro*-Michael reaction, incomplete aminolysis, intra- and intermolecular crosslinking) increases with the dendrimer generation. In this regard, we followed the concepts outlined earlier [28] for the synthesis of **G2** dendrimers based on *p-tert*-butylthiacalix[4]arene and the ester derivatives as precursors (**G1.5**). Temperature of the reaction mixture was kept below 30 °C for reducing the possibility of side reactions. The reagents were used in sufficient excess, and residual reagents were removed by a long repetitive series of azeotropic distillations.

At the first step of the synthesis, the **G1** PAMAM-calix-dendrimers were treated with the methyl acrylate. The reaction was carried out for 36 h. to ensure complete conversion of the precursor compounds (Figure 1). Unreacted amine groups were monitored by a ninhydrin test (1% ninhydrin solution in ethanol), which was negative after 36 h. Thus, the **G1.5-cone**, **G1.5-paco** and **G1.5-alt** compounds containing 16 ester groups were obtained in high yields. At the second step, the **G1.5** compounds reacted with a large excess of ethylenediamine. The reaction was monitored by ^1^H NMR spectroscopy. Thus, after 70 h, residual signals of protons of methyl ester fragment were observed in the ^1^H NMR spectra of the reaction mixture. Therefore, the reaction time was prolonged to 90 h, and the target **G2** dendrimers with a thiacalixarene core in *cone* (**G2-cone**), *partial cone* (**G2-paco**) and *1,3-alternate* (**G2-alt**) conformations were obtained in high yields (85–90%).

The synthesized compounds were characterized by physical methods, i.e., ^1^H and ^13^C NMR spectroscopy, FTIR spectroscopy and high-resolution electrospray ionization (HR ESI) mass spectrometry (Appendix A).

Figure 1 shows stacked ^1^H NMR spectra of the **G1.5-alt** and **G2-alt** compounds. In both cases, proton signals of the macrocycle assigned to the *tert*-butyl, oxymethylene and aromatic fragments appeared as singlets at 1.28, 4.12 and 7.60 ppm, respectively. The signals of the methylene protons of the hexylidene fragment appeared as a series of multiplets at 1.34, 1.49 and 1.56 ppm. The singlet of protons of the terminal methyl fragments of the **G1.5-alt** was located at 3.66 ppm. It completely disappeared in the spectrum of the **G2-alt** after the reaction with ethylenediamine. An intense triplet at 2.72 ppm in the spectrum of **G2-alt** was assigned to the protons of the methylene fragment at the terminal primary amino groups. The proton signals of the second branching methylene groups shifted upfield (2.36 ppm) against corresponding signals of the **G1.5-alt** (2.46 ppm) due to the conversion of the ester group into the amide group. Similar pattern was observed in the ^1^H NMR spectra of the **G2-cone** and **G2-paco**. Absent of the methyl ester group signals, the shift of the proton signals shift refers to the branching methylene groups alongside the general integral intensity ratio, signal multiplicity made it possible to conclude that the reaction was complete and no defects in structure of **G2** compounds were observed.

Additional analysis of the synthesized **G1.5** and **G2** compounds was carried out by the FTIR spectroscopy. A significant absorption band of the carbonyl group of the ester fragment appeared at 1734 cm^−1^ in the IR spectra of **G1.5-cone**, **G1.5-paco** and **G1.5-alt** compounds, alongside amide I and amide II bands at 1648 cm^−1^ and 1534 cm^−1^, correspondingly. In addition, bands of the N-H and C_Ph_-O-CH_2_ bonds appeared at 3318 cm^−1^ and 1086 cm^−1^. The aforementioned carbonyl group band completely disappeared in the IR spectra of the **G2** compounds, while an additional band appeared at 3061 cm^−1^, and the N-H bond band shifted to 3282 cm^−1^ and intensified. This is probably due to an increase in intra- and intermolecular H–bonding for the **G2** compounds compared to the **G1.5**. Absence of carbonyl group band in the IR spectra of the **G2** dendrimers confirms the completion of the reaction and the replacement of all terminal fragments by the amine groups.

HR ESI mass spectrometry data also confirmed the structure of the synthesized dendrimers. This ionization method is widely used to study the PAMAM dendrimers [31,32] because it does not cause destruction of the macromolecule and complication of the spectra, so that the peaks of the polycharged ions can be observed in the mass spectra. The peaks corresponding to the polyprotonated molecular ions with sodiumion ([M + 8H + Na]^9+^, [M + 11H + Na]^12+^, [M + 17H + Na]^18+^) were found in all the mass spectra of the **G2** dendrimers.

Thus, a convenient method has been developed for the synthesis of the second generation of the PAMAM-calix-dendrimers with cores in *cone*, *partial cone* and *1,3-alternate* conformations.

### 3.2. Spectral Properties of the First and Second Generation of PAMAM-Calix-Dendrimers

Today, nontraditional fluorophores are of great interest [33]. Usually, such compounds include siloxanes, polymers and nonaromatic peptides. Dendrimers also belong to the nontraditional fluorophores. For example, a weak fluorescence of the carboxylate-terminated poly(amidoamine) dendrimers was found in 2001 [34]. To date, there is no consensus on the reasons for this phenomenon; suggestions on the key factors explaining the ability of the PAMAM dendrimers’ ability to fluoresce have been proposed [35]. They include aggregation-induced emission (AIE), interactions between the chromophores, spatial features and oxidation of the compounds. An increase in the fluorescence intensity of dendrimers with an increasing number of generations, as well as in that in an acidic environment, was described [36,37,38]. In our research, fluorescence of the obtained PAMAM-calix-dendrimers was found.

The investigation of the synthesized PAMAM-calix-dendrimers revealed their ability to exhibit fluorescence in methanol and water. Interestingly, the highest fluorescence intensity both for the first (**G1**) and second generation (**G2**) of the dendrimers was observed for the *cone* conformation of a macrocyclic dendrimer core. The *partial cone* stereoisomers were inferior in emission intensity, while the dendrimers with a macrocyclic core in *1,3-alternate* conformation exerted weakest fluorescence (Figure 2a and Appendix A). It is important to note that the emission intensity of the **G2** dendrimers appeared to be higher than that of the **G1** dendrimers [37].

In the case of the protonated derivatives of dendrimers, a significant red shift of emission maxima was observed (Figure 2b,c, Appendix A). Emission maxima shifted from 390 to 399 (**G1-cone**) and from 400 to 435 nm (**G1-paco**) (Figure 2b and Appendix A); from 398 to 430 (**G2-cone**) and from 402 to 434 nm (**G2-paco**) (Appendix A) for *cone* and *partial cone* stereoisomers. For the *1,3-alternate* stereoisomer, other spectral changes were observed. In the case of the **G1-alt**, the emission maximum shifted from 380 to 368 nm. However, along with this, a long-wavelength maximum appeared at 430 nm (Figure 2c). In the case of the **G2-alt**, changes were similar, e.g., emission maximum shifted from 380 to 370 nm, and new a maximum appeared at 440 nm (Appendix A). We assume that the presence of macrocyclic platform aromatic fragments plays a key role in the synthesized dendrimers’ fluorescence. The fluorescence spectra of the nonmacrocyclic monomer of dendrite structure (**G1-monomer**) and classic first and second generation **G1** and **G2** PAMAM dendrimers (containing an ethylenediamine core) were recorded to confirm this hypothesis. The fluorescence of classic PAMAM dendrimers was extremely low (Appendix A). Meanwhile, the **G1-monomer** exhibited an intense fluorescence with two emission maxima (307 and 420 nm) in the spectra compared to those of the macrocyclic analogs **G1** (Appendix A). In our opinion, the fluorescence of dendrimers with a macrocyclic core is also related to the possibility of amine groups’ protonation in water. It leads to changes in the polarity of the macrocyclic core environment and to the red shift of emission maximum. Such changes were described in our previous research for the charged pillar[5]arene derivatives [39]. Another argument in favor of this hypothesis is that thiacalix[4]arenes do not usually exhibit fluorescent properties in the absence of fluorophore groups. However, fluorescence properties of phosphonate thiacalix[4]arene derivatives were recently investigated [40], revealing the increase in emission of these macrocycles upon conversion into charged (anionic) form. Along with this, the DLS experiments of the **G1-monomer** were performed, and the formation of monodisperse systems with particles size of 105-115 nm was observed (Table 1).

Protonated derivatives of the monomer compound (**G1-monomer∙HCl**) did not form stable colloid systems. It should be noted that the **G1-monomer∙HCl** emission intensity in the long-wavelength region of the spectrum (420 nm) was significantly lower than that of the **G1-monomer**, while the emission intensity in the short-wavelength region (307 nm) increased (Appendix A). This can be explained by the AIE effect, which is combined with the possibility of amino groups’ protonation of the dendrimer/monomer. The long-wavelength maximum in the spectrum can be attributed to the emission of the monomer excimer, which is formed due to π-π-stacking of aromatic fragments [41,42]. Protonation of the amino groups in the **G1-monomer∙HCl** leads to repulsion of positively charged groups, which resulted in the decrease in the intensity of the second emission maximum (420 nm). The polarity of the solvent has a significant influence on the fluorescence spectrum. While methanol is less polar than water (dielectric constant 78.36 for water vs. 32.66 for methanol) [43], an intense band at 307 nm and a less intense one at 410 nm were observed. This is in good accordance with the hypothesis of an excimer formation [44]. The emission band of the **G1-monomer∙HCl** at 410 nm completely disappeared in methanol. It should be noted that for **G1∙HCl** and **G2∙HCl**, the spectra in methanol emission maxima shifted at the long-wavelength region, similarly to what was observed in the aqueous solution (Figure 2b,c). Probably, the ionization of dendrimers becomes crucial for methanol solutions less polar than water [43]. These phenomena confirm our hypothesis on the nature of the PAMAM-calix-dendrimers’ fluorescence properties.

The DLS data also confirm the hypothesis of the AIE effect (Table 1, Appendix A). The most stable associates were obtained for **G1-monomer**, **G1-cone** and **G2-cone**. Their emission intensities were the highest among all the compounds.

The fluorescence intensity decreased with increasing polydispersity of the systems. Almost all the systems, except one (**G1-cone**), possessed a high (>0.25) polydispersity index (PDI) (Table 1). Apart from the model compound **G1-monomer** (PDI = 0.26), the **G1-cone** solution in the concentration of 1 × 10^−5^ M was relatively monodisperse (PDI = 0.21), with the particles size of 258 nm. All branches of the dendrimers with a macrocyclic core in *cone* conformation are brought as close as possible. This can also contribute to the AIE effect.

Thus, we suggest that the presence of the aromatic fragments of the macrocyclic core has a key influence on the luminescent properties of the PAMAM-calix-dendrimers. Along with that, association of the compounds and the ability of their amino groups to be protonated have an impact on luminescence, as well.

### 3.3. Binding of the Catecholamines by the G1 and G2 PAMAM-Calix-Dendrimers

Development of receptors and systems for targeted delivery of biologically active compounds is one of the most important applications of the PAMAM dendrimers. In this regard, we have investigated complexing properties of the synthesized dendrimers toward a series of catecholamines. As noted above, catecholamines are markers of many diseases of the central nervous system, and they can also be used as medicines [7,45,46]. Therefore, searching compounds capable of recognition and/or binding for the delivery of catecholamines is highly relevant.

Previously, the ability of the first generation of PAMAM-calix-dendrimers (**G1**) to bind salmon sperm DNA was shown [28,47]. However, cationic dendrimers are known to be inferior in their biocompatibility to anionic and neutral analogues [8,48] by causing a significant hemolysis. Therefore, additional protonation and conversion into an ammonium form of the synthesized **G1** and **G2** dendrimers were not performed in this study.

The investigated catecholamines, e.g., dopamine, *L*-noradrenaline (hereinafter noradrenaline) and *L*-adrenaline (hereinafter adrenaline), were used in the forms of hydrochlorides (Figure 3) to minimize the oxidation of catechol fragment to quinoid structure [49]. Initially, the UV-Vis spectroscopy was applied to investigate the ability of the **G1** and **G2** to bind catecholamines. Dopamine, noradrenaline and adrenaline have similar UV-Vis spectra, with the absorption maxima at 203, 220 and 280 nm corresponding to the π-π* transitions of the aromatic fragments [50]. The synthesized **G1** and **G2** dendrimers also exhibit absorption at 200–300 nm due to the π-π* transitions of the aromatic fragments of the macrocyclic core and the n-π* transitions of the carbonyl groups of dendrons. In the case of dendrimers with macrocyclic cores in *1,3-alternate* conformation, an additional absorption band at 255 nm was observed. This band is the spectral characteristic of this conformation in contrast to *cone* and *partial cone* [51,52]. The presence of this absorption band results from the different spatial positions of the chromophore fragments against each other and from the different strengths of their intramolecular interactions compared to those of other stereoisomers [52].

In the presence of the PAMAM-calix-dendrimers, the biggest changes in the absorption spectra were observed for dopamine. In addition to the hypochromic effect at 203 nm, common to other catecholamines, a remarkable hypochromic effect was also observed at 280 nm (Figure 4 and Appendix A).

However, quantification of the interaction of catecholamines with dendrimers by the UV-Vis spectroscopy method was rather difficult due to the overlapping of the absorption bands of the compounds studied. It is known that catecholamines contain a fluorophore fragment, which causes intense emission with a maximum at 316 nm when excited by light at 280 nm [53]. The calculation of binding constants by fluorescence spectroscopy is possible due to the absence of dendrimer fluorescence in this region.

A significant decrease in emission intensity with increasing dendrimer concentration was found in the fluorescence spectra of catecholamines with different concentrations of the **G1** and **G2** (Figure 5, Appendix A). Thus, the binding ability of all the synthesized dendrimers toward all the studied catecholamines was confirmed. Additional experiments with the **G1-monomer** were carried out to establish the role of the macrocyclic core in the catecholamines’ binding. There were no noticeable changes from additively derived absorption spectra after mixing **G1-monomer** with catecholamines. This indicated an absence of the interactions between the components in the mixtures (Appendix A). Furthermore, the fluorescence intensity of catecholamines was not changed, either, in the presence of the **G1-monomer** (Appendix A), confirming the key role of the macrocyclic platform in the binding hormones studied. Clearly, the fixation of dendrons to the thiacalixarene platform contributes to the multivalency effect in binding, which ultimately causes catecholamine binding [25,54,55].

The binding constants were calculated based on the data obtained from the fluorescence titration. Mathematical data processing was carried out using the Bindfit application, widely used in supramolecular chemistry [56,57,58]. A binding model with the host/guest ratio of 1:1 was selected (Table 2 and Appendix A), because calculation of the results in this model resulted in minimal errors.

The binding constants data made it possible to estimate the effect of a dendrimer generation and of the conformation of a macrocyclic core on binding efficiency toward the catecholamines. Among the first and second generation dendrimers, the highest binding constants were found for the **G1-alt** and **G2-alt**. This is probably due to their spatial structure, specifically, symmetrical arrangement of the dendrons relative to the macrocyclic platform. Such an arrangement provides free spatial positioning for dendrons and maximizes access to the binding sites. The values of binding constants for the dendrimers with the macrocyclic core in *cone* and *partial cone* conformations were close to each other. It should be noted that there was not significant selectivity in binding toward the catecholamines, despite a relatively high binding efficiency (logK_a_ = 3.85–4.72). For the **G2** dendrimers, the dependence of the catecholamine binding constants on the macrocyclic core conformation of the dendrimer was negligible. Meanwhile, binding constants of the **G2** were close to those of the **G1**. Hence, no significant effect of the size and number of generations of the investigated PAMAM-calix-dendrimers on the efficiency of binding was observed for the catecholamines studied. However, additional hydroxyl groups in the structure of the catecholamines contribute to binding to dendrimers by the formation of additional H-bonds, even without significant selectivity. Thus, the dopamine binding constants were the lowest ones, probably due to the absence of the OH- group in the alkyl fragment of the molecule. Interestingly, presence of the methyl group in the catecholamine structure contributes to the binding, as well. For the first generation of the PAMAM-calix-dendrimers, highest constants were observed in case of noradrenaline, whereas those of the second generation were observed for adrenaline.

The ^1^H and ^1^H-^1^H NOESY NMR spectroscopy were applied to establish the structure of the complexes formed and the contribution of methyl group of catecholamine to the complexing properties. Based on the fluorescent titration, the **G1-alt** and **G2-alt** compounds were chosen for the NMR experiments due to their highest binding constants toward catecholamines. The ratio was 1:1 based on the established stoichiometry of complexation (Figure 6, Appendix A).

The ^1^H NMR spectra of the mixtures of the **G1-alt** or **G2-alt** with the catecholamines (1 × 10^−2^ M, ratio 1:1, D_2_O, 25 °C, 400 MHz) were recorded. The upfield shift of both aromatic and aliphatic protons of the catecholamines was observed in all spectra of the **G1-alt**/catecholamine mixtures. This is due to the formation of H-bonds between the hydroxyl and amine groups of catecholamines and the amide and amino groups of the dendrimer macromolecules, alongside with the shielding of catecholamine by the amide groups of the dendrimer, caused by a substrate implemented into internal cavities of the dendrimer. A similar pattern was observed for the ^1^H NMR spectra of the **G2-alt**/catecholamines mixtures. The catecholamine proton signals were shifted upfield to a considerably greater extent than those of the **G1-alt**/catecholamine mixtures due to the increased number of the amide groups in the dendrimer structure that led to the shielding enhancement.

Figure 6 shows the adrenaline ^1^H NMR spectrum against that of its mixtures with **G1-alt** and **G2-alt** to demonstrate the aforementioned phenomenon. The difference in the chemical shift changes of catecholamine proton signals for the **G1-alt**/catecholamine and **G2-alt**/catecholamine mixtures correlated with the changes in the logarithms of the binding constants calculated for **G1-alt** and **G2-alt**.

However, the ^1^H NMR spectroscopy was not suitable for the establishment of the complex structure and determination of the dendrimer groups involved in complexation. Theoretically, dendrimers are able to bind substrates by their terminal groups as well as by inclusion in the internal cavity. ^1^H-^1^H NOESY NMR spectroscopy was applied to additional investigation of complexes (Figure 7 and Appendix A). Previously, we have found for linear lactide derivatives of thiacalix[4]arene [29] that catecholamines as the guest molecules were oriented toward the macrocycle by either an aromatic or aliphatic fragment depending on the nature of the catecholamine. Herein, cross-peaks between the protons of the aliphatic catecholamine fragment and the protons **H^3^**, **H^4^** of the dendrimer hexylidene fragment were observed in the 2D ^1^H-^1^H NOESY NMR spectra of the **G1-alt**/catecholamine mixtures. Meanwhile, there were no cross-peaks between the protons of the catecholamine aromatic fragment and of the dendrimer. Figure 7a shows the 2D ^1^H-^1^H NOESY NMR spectrum of the **G1-alt**/noradrenaline mixture. The cross-peaks between the methylene protons of noradrenaline **H^c^** and the protons **H^3^**, **H^4^** of the dendrimer hexylidene fragment are observed. In the case of the **G2-alt**/catecholamine 2D ^1^H-^1^H NOESY NMR spectra, identical cross-peaks between the **H^c^** catecholamine protons and the **H^3^**, **H^4^** dendrimer protons were recorded with additional cross-peaks between the **H^c^** protons and nearby protons of the dendrimer hexylidene fragment **H^2^**, **H^5^**. This is due to the enlargement of a dendrimer internal cavity caused by the steric effects of second-generation dendrons. Thus, cross-peaks between aliphatic adrenaline protons **H^c^** and dendrimer protons **H^2^**, **H^3^**, **H^4^**, **H^5^**, alongside with ones between protons **H^d^** and **H^2^**, **H^5^**, were observed in the 2D ^1^H-^1^H NOESY NMR spectrum of the **G2-alt**/adrenaline mixture (Figure 7b). No other significant cross-peaks were found in the **G2-alt**/catecholamine spectra. Hence, the position of catecholamines in the complexes remained unchanged compared to that in the **G1-alt**/catecholamine complexes.

Differences in the catecholamine binding constants can be explained by the 2D ^1^H-^1^H NOESY NMR spectroscopic data. In the case of the **G1** PAMAM-calix-dendrimers, maximal binding constants were observed for noradrenaline, as well as in the case of the **G2** for adrenaline. Probably, this is due to the structural difference between the substrates (adrenaline molecule has a methyl group at the nitrogen atom), as well as due to the difference in internal cavities size between the **G1** and **G2** PAMAM-calix-dendrimers. Thus, the adrenaline-bearing methyl group requires more space in the dendrimer internal cavities for effective binding, which is possible only in case of the **G2** dendrimers. Noradrenaline, which does not have a methyl group, binds more efficiently to the **G1** dendrimers.

The 2D ^1^H-^1^H NOESY NMR data for the **G1-alt**/catecholamines and **G2-alt**/ catecholamines complexes indicated an orientation of all the guest molecules by their aliphatic fragment put into the thiacalixarene dendrimer core. Furthermore, the complexation does not take place via terminal groups of the dendrimer but within the pseudo-cavity formed by the macrocycle dendritic substituents. Therefore, change in the number of terminal functional groups leads only to the enlargement of the dendrimer internal cavity size, causing increased adrenaline binding efficiency for the **G2** PAMAM-calix-dendrimers. Thus, H-bonding and van der Waals forces make a key contribution to the interaction of PAMAM-calix-dendrimers with catecholamines. The location of hormones relative to the macrocyclic fragment indicates the almost complete absence of hydrophobic interactions inherent in catecholamines [59].

The sizes of the **G1** and **G2** complexes with catecholamines were investigated by the DLS method (Appendix A). Upon the addition of catecholamines to the **G1-cone** and **G2-cone** compounds, a minor decrease in PDI was observed against that of the individual dendrimers (Table 1 and Table 3). Thus, all three systems with catecholamines were monodispersed for the **G1-cone** (PDI = 0.22–0.25). However, associate sizes were too large (376 nm and 516 nm for dopamine and noradrenaline, respectively), except that for the **G1-cone**/adrenaline system (273 nm). It should be noted that the **G1-cone** formed monodisperse systems with the size up to 258 nm for all the concentrations studied, even in the absence of catecholamines. In case of the **G2-cone**, not only PDI but also the size of associates formed in presence of adrenaline decreased against individual dendrimer (311 vs. 423 nm). Thus, a spatial convergence of substituents in the **G2** dendrimer was observed in the presence of catecholamine compared to the individual dendrimer. The formation of the monodisperse systems with rather small particle sizes for the **G1-cone** and **G2-cone** is probably due to the amphiphilic structure of these dendrimers, in which hydrophilic dendrons and the hydrophobic macrocyclic core are spatially separated. The **G1-paco**/catecholamines systems were also quite monodisperse (PDI 0.20–0.30), but the size of associates was significantly larger (536–745 nm). The **G2-paco**/catecholamines systems exhibited a high polydispersity (PDI 0.50–0.63). Clearly, symmetric structures of the **G1-alt** and **G2-alt** have the most noticeable difference from asymmetrical structures of the **G1-cone** and **G2-cone**. Thus, most polydisperse systems (PDI 0.49–0.89) were observed for the complexes of the **G1-alt** and **G2-alt** with catecholamines. In them, the associate size was in a micron scale, with one exception of the **G1-alt**/adrenaline system (PDI = 0.25, average size of 654 nm). The **G1-alt** and **G2-alt** formed large associates, even in the absence of catecholamines (903–1208 nm). Probably, in the case of dendrimers with the core in *cone* and *partial cone* conformation, the formation of long form particles is possible [60] due to hydrophobic interactions between the macrocyclic fragments. These interactions lead to the enlargement of the particles formed in the system.

As a result, the ability of the **G1** and **G2** PAMAM-calix-dendrimers to effectively bind dopamine, *L*-adrenaline and *L*-noradrenaline (logK_a_ 3.85–4.75) has been established by spectral methods. The highest binding constants were observed for the dendrimers with macrocyclic cores in *1,3-alternate* conformation. Presence of the hydroxyl group in the aliphatic catecholamine fragment increased the binding constant. It was shown by the 2D ^1^H–^1^H NOESY NMR spectroscopy that the complexation occurred within the pseudo-cavity formed by the macrocycle hexylidene substituents. The formation of the monodisperse systems in case of the **G1-cone** and **G2-cone** dendrimers along with the **G1-paco** has been established by the DLS method. The **G1-cone** dendrimer formed with dopamine and adrenaline associates with the size of 376 and 273 nm, respectively.

The study of the resulting supramolecular systems using transmission electron microscopy (TEM) confirms the formation of submicron associates with catecholamines. Appendix A show TEM images of the self-assembly of **G1-cone** and **G2-cone** compounds, as well as **G1-cone**/dopamine, **G1-cone**/adrenaline, **G1-cone**/noradrenaline, and **G2-cone**/dopamine associates as the most monodisperse systems. Amorphous nanoparticles with high polydispersity are formed as a result of self-assembly of the **G2-cone** compound, which was also previously established for the **G1-cone** compound [28]. The comparison of TEM images of the **G1-cone**/dopamine, **G1-cone**/adrenaline and **G1-cone**/noradrenaline supramolecular systems showed that nanoparticles with the size close to 80–100 nm coalesced with each other and are formed (Appendix A) regardless of catecholamine bound. The size of the resulting dopamine associates, i.e., **G1-cone**/dopamine and **G2-cone**/dopamine, does not change significantly (Appendix A) with an increase in the generation of PAMAM-calix-dendrimers (**G1-cone** and **G2-cone**). The data obtained are in good agreement with the correlations obtained using DLS. The larger size of associates obtained using the DLS method is explained by the fact that this method shows the sizes of associates with a solvate shell, in contrast to electron microscopy.

Further elemental mapping of the dendrite structure forming as a result of coalescence of nanosized **G2-cone**/dopamine associates was carried out (Appendix A) by energy dispersive analysis. Elemental maps for S, N, Cl, C atoms were registered. A correlation was established between the locations of these elements, i.e., the areas of these elements intersect and overlap each other by comparing the spatial features on the S, N, Cl, C maps. Thus, it was proved that the obtained TEM images correspond to associates formed by PAMAM-calix-dendrimers (elements S, N, C) and dopamine hydrochloride (N, Cl, C).

### 3.4. Hemolytic Activity of the G1-alt and G2-alt PAMAM-Calix-Dendrimers

The toxicity level of new compounds to living organisms is an important factor for their application in various fields, especially in biomedicine. Because it was established in Section 3.3 that catecholamines are most effectively bound by the PAMAM-calix-dendrimers in *1,3-alternate* conformation with the most freely spaced branches, the hemolytic activity of the compounds **G1-alt** and **G2-alt** was further evaluated. Figure 8 presents data on the effect of the compounds **G1-alt** and **G2-alt** on RBC. The concentrations, i.e., 20 μM for the **G1-alt** and 10 μM for the **G2-alt**, were chosen on the base of the considerations of charge–charge ratios, which depended on the surface charge of the studied dendrimers. After 3 h. of incubation, only the **G1-alt** showed significant effect on hemolysis, while the **G2-alt** caused minor toxicity. The level of hemolysis significantly increased after 24 h. of incubation, but the trend of decreasing toxicity with increasing generation remained. The toxicity of classical PAMAM dendrimers increases with the number of generation [19,61]. However, this is not always observed and depends on the structure, number and lability of the branches [62]. In our case, an inverse correlation of the generation number vs. toxic effect took place. It should be assumed that the thiacalix[4]arene core and the emerging features of the lability of the dendrimer branches play a significant role in the mechanism of toxicity.

### 3.5. In Vitro Release Studies of Catecholamines from Their Complexes with PAMAM-Calix-Dendrimers

The identification of the effect of complex formation with PAMAM-calix-dendrimers on the release of catecholamines is of considerable interest. Dendrimers (especially high generations) are able to slow down the release of drugs upon their binding [18,63]. Slowing the release of catecholamines may be useful in the development of prolonged action drugs. The study was carried out on the **G2-cone** compound, which successfully combines both high-binding constants and relatively small sizes of associates. It turned out that the complex formation with this dendrimer slightly slows down the release of all studied catecholamines compared to catecholamine solutions (Appendix A). It is interesting to note that a comparison of the fluorescence spectra of the retentate and dialysate after 24 h showed a significant difference between them for different systems. The spectra of dialysate and retentate are identical (Appendix A) in the case of catecholamine solutions. Only catecholamines are present in the dialysate for their mixtures with **G2-cone**, while the retentate is enriched in dendrimer (Appendix A). A significant decrease in the emission of catecholamines in the mixture retentate compared to the dialysate is explained by quenching due to the binding.

## 4. Conclusions

A simple synthetic method for the first and second generation PAMAM-calix-dendrimers with *p-tert-*butylthiacalix[4]arene in *cone*, *partial cone* and *1,3-alternate* conformations as the core was developed. It consisted of sequential reactions with methyl acrylate and ethylenediamine. The obtained PAMAM-calix-dendrimers were found to be fluorescently active. The fluorescence properties of the compounds are due to the presence of the aromatic macrocyclic platform of thiacalixarene in combination with the possibility of the AIE effect and the protonation of amine groups in protogenic solvents. The ability of the obtained **G1** and **G2** PAMAM-calix-dendrimers to effectively bind catecholamines (dopamine, *L*-adrenaline and *L*-noradrenaline) was established by spectral methods. The formation of the smallest and most monodisperse particles (273–516 nm, PDI 0.22–0.25) with all the catecholamines by the **G1-cone** dendrimer was shown by the DLS method. Binding substrates by dendrimers occurred within the internal cavity formed by the macrocycle substituents, as was established by the 2D ^1^H–^1^H NOESY NMR spectroscopy. The efficiency of catecholamine binding increased in the presence of hydroxyl groups in their structure, i.e., dopamine was bound by minimal efficiency. The methyl group in the aliphatic fragment of the catecholamine also affected binding by PAMAM-calix-dendrimers, i.e., bulkier adrenaline interacted more efficiently with the **G2** dendrimers, which had a less dense packing of amidoamine substituents. Study of the PAMAM-calix-dendrimers’ hemolytic activity showed its decrease in the range from the first to the second generation, which favorably distinguished them from classical PAMAM dendrimers. Thus, the prospects of binding catecholamines by the first and second generation dendrimers with a thiacalixarene core were revealed. They can be demanded in the development of low-cost drug delivery systems. However, our work is the first step toward this goal. In future works, we will pay close attention to the pharmacological activity, pharmacokinetics and biodistribution of the PAMAM-calix-dendrimers.

## Data Availability

The data presented in this study are available in Appendix A.

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
