# Peer review of "PAMAM-Calix-Dendrimers: Second Generation Synthesis, Fluorescent Properties and Catecholamines Binding"

_pharmaceutics, 2022, doi:10.3390/pharmaceutics14122748_

Round 1

Reviewer 1 Report

This work included the development of synthetic methods for second-generation dendrimers based on thiacalix[4]arene and their application for catecholamines binding. The authors show the relationship between the dendrimers generation and the catecholamine binding efficiency. experientially the authors do a good job determining the structure of the resulting complexes using a series of physical methods. Overall their results show the potential application of the synthesized compound to bind catecholamines. In particular, the first and second-generation dendrimers with a thiacalixarene core.

A few suggestions to improve the paper are:

1) minor editing is required for spelling and grammar. 

2) a little more detail describing the type of binding between the drugs and the dendrimers can be useful

3) if these complexes are to be used as drugs information about how the drugs are released from the dendrimer complexes would be essential

Author Response

First of all, we would like to thank respected Reviewer for careful consideration of the manuscript. In accordance with the comments, the following changes have been made:

1) minor editing is required for spelling and grammar.

Answer:

The manuscript was checked and significant changes were made in spelling and grammar.

2) a little more detail describing the type of binding between the drugs and the dendrimers can be useful

Answer:

The following part was added to the manuscript discussing the complexation of drugs with dendrimers. Relevant references were also added.

«The dendrimer platform is convenient for binding drug molecules. These molecules can both be physically distributed inside the cavities of dendrimers and can bind to it through covalent bonds and a whole range of non-covalent interactions (electrostatic interactions, hydrogen bonding and van der Waals forces) [12,18].»

3) if these complexes are to be used as drugs information about how the drugs are released from the dendrimer complexes would be essential

Answer:

We have studied the release of catecholamines from complexes with dendrimers. The corresponding fragment has been added to the manuscript and electronic supplementary materials.

«3.5. In vitro release studies of catecholamines from their complexes with PAMAM-calix-dendrimers

The identification of the effect of complex formation with PAMAM-calix-dendrimers on the release of catecholamines is of considerable interest. Dendrimers (especially high generations) are able to slow down the release of drugs upon their binding [18, 63]. Slowing the release of catecholamines may be useful in the development of prolonged action drugs. The study was carried out on the G2-cone compound, which successfully combines both high binding constants and relatively small sizes of associates. It turned out that the complex formation with this dendrimer slightly slows down the release of all studied catecholamines compared to catecholamine solutions (Figures S117-S134). It is interesting to note that a comparison of the fluorescence spectra of the retentate and dialysate after 24 hours showed a significant difference between them for different systems. The spectra of dialysate and retentate are identical (Figures S129-S131) in the case of catecholamine solutions. Only catecholamines are present in the dialysate for their mixtures with G2-cone, while the retentate is enriched in dendrimer (Figures S132-S134). A significant decrease in the emission of catecholamines in the mixture retentate compared to the dialysate is explained by quenching due to the binding.»

Reviewer 2 Report

In this article, Mostovaya et al. describe the synthesis of PAMAM-thiacalixarene-based dendrimers and their use as binding platforms for catecholamines. After describing the synthesis of the dendrimers, the authors study their binding efficacy to three different neurotransmitters: dopamine (DA), adrenaline (Adr), and nor-adrenaline (NAdr) by spectroscopic techniques (NMR and UV-vis). They also briefly explore the toxicity of the objects based on hemolysis tests.

The article is written in rather clumsy English and should definitely be checked by a native speaker.

There are many weaknesses to this paper:

-    The introduction does not justify why the authors want to use dendrimers to bind catecholamines. How do they plan to use the objects for drug delivery? Clear justification should be given as to why anyone would like to encapsulate/bind water-soluble molecules… The authors state “A significant increase in life expectancy has already led to an increase in neurodegenerative diseases (about 10 million new cases of dementia each year) [5, 6]. The catecholamines application in anesthetic practice as vasopressors and inotropes is also widely known [7]. So, the importance and relevance of the design of compounds capable of binding catecholamines becomes clear.” Which is simply not convincing at all.

-    The authors claim that their dendrimers have a size comparable to the size of “biological molecules”. What does that mean? There are so many different “biological molecules” with very different sizes. A statement like this is meaningless on its own.

-          What’s the relevance of transporting water-soluble molecules in aggregates of macromolecules? Do the authors believe that these aggregates (d ranging from 400 nm to 1400 nm) will have favorable PK/PD properties? Why and how?

-          The nature of the catechomamine-dendrimer interaction is not explained well-enough and UV-vis/NMR studies are not sufficient.

-          Catecholamine release studies should be carried out, maybe by dialysis vs serum, for instance…?

-          The characterization of the particles/aggregates is also lacking. The objects are huge and very polysiperse. What’s their morphology? TEM/SEM images should be provided. It might be interesting to do EDS mapping of the aggregates in order to see how nitrogen is distributed in the aggregates.

-          Nothing is said about the biological activity of the catecholamine-loaded dendrimers neither on cells nor animals.

I honestly do not believe that this article is solid enough for publication in Pharmaceutics. All we learn is that catecholamines seem to bind (to some extent) to the dendrimers. There is no mention of the pharmacological activity, pharmacokinetics or biodistribution of the objects, all of which seem essential in the “drug-delivery” perspective.

Author Response

First of all, we would like to thank respected Reviewer for careful consideration of the manuscript. In accordance with the comments, the following changes have been made:

  1. The article is written in rather clumsy English and should definitely be checked by a native speaker.

Answer:

The manuscript was checked and significant changes were made in spelling and grammar.

  1. The introduction does not justify why the authors want to use dendrimers to bind catecholamines. How do they plan to use the objects for drug delivery? Clear justification should be given as to why anyone would like to encapsulate/bind water-soluble molecules… The authors state “A significant increase in life expectancy has already led to an increase in neurodegenerative diseases (about 10 million new cases of dementia each year) [5, 6]. The catecholamines application in anesthetic practice as vasopressors and inotropes is also widely known [7]. So, the importance and relevance of the design of compounds capable of binding catecholamines becomes clear.” Which is simply not convincing at all.

Answer:

The Introduction has been rewritten according the comments of all Reviewers. The following has been added.

«The reversible binding of catecholamines may be useful for the creation of sustained-release formulations.»

  1. The authors claim that their dendrimers have a size comparable to the size of “biological molecules”. What does that mean? There are so many different “biological molecules” with very different sizes. A statement like this is meaningless on its own.

Answer:

The text has been rewritten, the comparison with biomolecules has been removed.

«The nanometer size of dendrimer molecules comparable to that of the drug delivery systems facilitates their application in medical technologies [12].»

  1. What’s the relevance of transporting water-soluble molecules in aggregates of macromolecules? Do the authors believe that these aggregates (d ranging from 400 nm to 1400 nm) will have favorable PK/PD properties? Why and how?

Answer:

The following fragment has been added to the manuscript:

«Dendrimers (especially high generations) are able to slow down the release of drugs upon their binding [18, 63]. Slowing the release of catecholamines may be useful in the development of prolonged action drugs. »

Pharmacokinetics and pharmacodynamics properties study was not the aim of this work. However, it is undoubtedly necessary to continue the work in which the first but promising steps have been taken by now. The study of the catecholamines release, which was proposed by the Reviewers, showed the possibility of using obtained dendrimers to prolong the action of drugs. A discussion of these results has been added to the manuscript. We will pay the greatest attention to this problem in our future publications.

  1. The nature of the catechomamine-dendrimer interaction is not explained well-enough and UV-vis/NMR studies are not sufficient.

Answer:

A discussion of the binding nature of catecholamines by dendrimers has been added to the text.

«Thus, H-bonding and van der Waals forces make a key contribution to the interaction of PAMAM-calix-dendrimers with catecholamines. The location of hormones relative to the macrocyclic fragment indicates the almost complete absence of hydrophobic interactions inherent in catecholamines [59].»

  1. Catecholamine release studies should be carried out, maybe by dialysis vs serum, for instance…?

Answer:

We have studied the release of catecholamines from complexes with dendrimers. The corresponding fragment has been added to the manuscript and electronic additional materials.

«3.5. In vitro release studies of catecholamines from their complexes with PAMAM-calix-dendrimers

The identification of the effect of complex formation with PAMAM-calix-dendrimers on the release of catecholamines is of considerable interest. Dendrimers (especially high generations) are able to slow down the release of drugs upon their binding [18, 63]. Slowing the release of catecholamines may be useful in the development of prolonged action drugs. The study was carried out on the G2-cone compound, which successfully combines both high binding constants and relatively small sizes of associates. It turned out that the complex formation with this dendrimer slightly slows down the release of all studied catecholamines compared to catecholamine solutions (Figures S117-S134). It is interesting to note that a comparison of the fluorescence spectra of the retentate and dialysate after 24 hours showed a significant difference between them for different systems. The spectra of dialysate and retentate are identical (Figures S129-S131) in the case of catecholamine solutions. Only catecholamines are present in the dialysate for their mixtures with G2-cone, while the retentate is enriched in dendrimer (Figures S132-S134). A significant decrease in the emission of catecholamines in the mixture retentate compared to the dialysate is explained by quenching due to the binding.»

  1. The characterization of the particles/aggregates is also lacking. The objects are huge and very polysiperse. What’s their morphology? TEM/SEM images should be provided. It might be interesting to do EDS mapping of the aggregates in order to see how nitrogen is distributed in the aggregates.

Answer:

The samples were studied by TEM. A significant difference between dendrimers in the free state and when bound to catecholamines was shown. The mapping of the sample showed the simultaneous presence in the aggregates both sulfur (present only in the dendrimer) and nitrogen (present in dendrimer and catecholamines) with chlorine (present only in catecholamine chloride). The absence of areas that do not contain sulfur indicates an equal distribution of catecholamine/dendrimer and, accordingly, the formation of their complexes. The corresponding fragment has been added to the manuscript.

«The study of the resulting supramolecular systems using transmission electron microscopy (TEM) confirms the formation of submicron associates with catecholamines. Figures S108–109 show TEM images of the self-assembly of G1-cone and G2-cone compounds, as well as G1-cone/dopamine, G1-cone/adrenaline, G1-cone/noradrenaline, and G2-cone/dopamine associates as the most monodisperse systems. Amorphous nanoparticles with high polydispersity are formed as a result of self-assembly of the G2-cone compound, which was also previously established for the G1-cone compound [28]. The comparison of TEM images of the G1-cone/dopamine, G1-cone/adrenaline, and G1-cone/noradrenaline supramolecular systems was shown that nanoparticles with the size close to 80–100 nm coalesced with each other are formed (Figure S108) regardless of catecholamine bound. The size of the resulting dopamine associates, i.e., G1-cone/dopamine and G2-cone/dopamine, does not change significantly (Figure S109) with an increase in the generation of PAMAM-calix-dendrimers (G1-cone and G2-cone). The data obtained are in good agreement with the correlations obtained using DLS. The larger size of associates obtained using the DLS method is explained by the fact that this method shows the sizes of associates with a solvate shell, in contrast to electron microscopy. Further elemental mapping of the dendrite structure forming as a result of coalescence of nanosized G2-cone/dopamine associates was carried out (Figure S110–116) by energy dispersive analysis. Elemental maps for S, N, Cl, C atoms were registered. A correlation was established between the locations of these elements, i.e., the areas of these elements intersect and overlap each other by comparing the spatial features on the S, N, Cl, C maps. Thus, it was proved that the obtained TEM images correspond to associates formed by PAMAM-calix-dendrimers (elements S, N, C) and dopamine hydrochloride (N, Cl, C).»

  1. Nothing is said about the biological activity of the catecholamine-loaded dendrimers neither on cells nor animals.

Answer:

The biological activity of the catecholamine-loaded dendrimer study was not the goal of this rather large work. We believe that the results presented in this manuscript may provide a stimulus for the study of such compounds. We will undoubtedly pay close attention to this problem in future publications. A fragment reflecting these plans has been added to the Conclusion of the manuscript.

  1. I honestly do not believe that this article is solid enough for publication in Pharmaceutics. All we learn is that catecholamines seem to bind (to some extent) to the dendrimers. There is no mention of the pharmacological activity, pharmacokinetics or biodistribution of the objects, all of which seem essential in the “drug-delivery” perspective.

Answer:

We are glad that the Reviewer provide insight the work and showed such a high interest in it. We believe that the material obtained can cause discussion, which will contribute to the further development of the dendrimers chemistry. Undoubtedly, we will to continue the work and pay close attention to all the issues raised by the Reviewer. An excerpt has been added to the conclusion that addresses these issues.

«However, our work is the first step towards this goal. In future works, we will pay close attention to the pharmacological activity, pharmacokinetics and biodistribution of the PAMAM-calix-dendrimers.»

Reviewer 3 Report

In this manuscript, the authors synthesized the first and second generation PAMAM-calix-dendrimers based on p-tert-butylthiacalix[4]arene core in cone, partial cone and 1,3-alternate conformations. The obtained PAMAM-calix-dendrimers have effective bind ability to the catecholamines (dopamine, L-adrenaline and L-noradrenaline), and have application prospects in drug delivery. From my perspective, this paper is very meaningful, but should be revised and improved before considering to be published.    

1. The drawing of chemical structures in Scheme 1 are not normative, and some figures lack integrity and aesthetics, for example Fig. S31 and Fig. S33.

2. Page 10: “For cone and partial cone stereoisomers emission maxima shifted from 390 to 399 and from 400 to 435 nm (Figure 2a)”, whether figure and text do not correspond.

3. PAMAM has been widely studied in the field of drug delivery, it is better to demonstrate the advantages of this work compared with the previously reported PAMAM dendrimer molecules.

4. Please supplement the data of Transmission Electron Microscopy (TEM) to demonstrate the formation of particle with catecholamines by the dendrimers.

Author Response

First of all, we would like to thank respected Reviewer for careful consideration of the manuscript. In accordance with the comments, the following changes have been made:

  1. The drawing of chemical structures in Scheme 1 are not normative, and some figures lack integrity and aesthetics, for example Fig. S31 and Fig. S33.

Answer:

The diagram and figures have been corrected.

  1. Page 10: “For cone and partial cone stereoisomers emission maxima shifted from 390 to 399 and from 400 to 435 nm (Figure 2a)”, whether figure and text do not correspond.

Answer:

Unfortunately, we made a mistake. It has been corrected. The text fragment has been rewritten more clearly.

«Emission maxima shifted from 390 to 399 (G1-cone) and from 400 to 435 nm (G1-paco) (Figures 2b, S32); from 398 to 430 (G2-cone) and from 402 to 434 nm (G2-paco) (Figure S31) for cone and partial cone stereoisomers.»

  1. PAMAM has been widely studied in the field of drug delivery, it is better to demonstrate the advantages of this work compared with the previously reported PAMAM dendrimer molecules.

Answer:

We have added the following text to the Introduction of the manuscript:

«It is interesting that there are practically no examples in the literature of the use of PAMAM dendrimers for catecholamine binding. A dendrimer containing β-cyclodextrin fragments as terminal groups has been obtained quite recently [27]. The authors propose it as a dopamine sensor, but do not address the issues raising in current manuscript.»

  1. Please supplement the data of Transmission Electron Microscopy (TEM) to demonstrate the formation of particle with catecholamines by the dendrimers.

Answer:

The samples were studied by TEM. A significant difference between dendrimers in the free state and when bound to catecholamines was shown. The mapping of the sample showed the simultaneous presence in the aggregates both sulfur (present only in the dendrimer) and nitrogen (present in dendrimer and catecholamines) with chlorine (present only in catecholamine chloride). The absence of areas that do not contain sulfur indicates an equal distribution of catecholamine/dendrimer and, accordingly, the formation of their complexes. The corresponding fragment has been added to the manuscript.

«The study of the resulting supramolecular systems using transmission electron microscopy (TEM) confirms the formation of submicron associates with catecholamines. Figures S108–109 show TEM images of the self-assembly of G1-cone and G2-cone compounds, as well as G1-cone/dopamine, G1-cone/adrenaline, G1-cone/noradrenaline, and G2-cone/dopamine associates as the most monodisperse systems. Amorphous nanoparticles with high polydispersity are formed as a result of self-assembly of the G2-cone compound, which was also previously established for the G1-cone compound [28]. The comparison of TEM images of the G1-cone/dopamine, G1-cone/adrenaline, and G1-cone/noradrenaline supramolecular systems was shown that nanoparticles with the size close to 80–100 nm coalesced with each other are formed (Figure S108) regardless of catecholamine bound. The size of the resulting dopamine associates, i.e., G1-cone/dopamine and G2-cone/dopamine, does not change significantly (Figure S109) with an increase in the generation of PAMAM-calix-dendrimers (G1-cone and G2-cone). The data obtained are in good agreement with the correlations obtained using DLS. The larger size of associates obtained using the DLS method is explained by the fact that this method shows the sizes of associates with a solvate shell, in contrast to electron microscopy. Further elemental mapping of the dendrite structure forming as a result of coalescence of nanosized G2-cone/dopamine associates was carried out (Figure S110–116) by energy dispersive analysis. Elemental maps for S, N, Cl, C atoms were registered. A correlation was established between the locations of these elements, i.e., the areas of these elements intersect and overlap each other by comparing the spatial features on the S, N, Cl, C maps. Thus, it was proved that the obtained TEM images correspond to associates formed by PAMAM-calix-dendrimers (elements S, N, C) and dopamine hydrochloride (N, Cl, C).»

Round 2

Reviewer 2 Report

The authors have thouroughly improved their article. It can now be considered for publication.